# CONTRASTIVE REPRESENTATIONS MAKE PLANNING EASY

## ABSTRACT

Probabilistic inference over time series data is challenging when observations are high-dimensional. In this paper, we show how inference questions relating to prediction and planning can have compact, closed form solutions in terms of learned representations. The key idea is to apply a variant of contrastive learning to time series data. Prior work already shows that the representations learned by contrastive learning encode a probability ratio. By first extending this analysis to show that the marginal distribution over representations is Gaussian, we can then prove that conditional distribution of future representations is also Gaussian. Taken together, these results show that a variant of temporal contrastive learning results in representations distributed according to a Gaussian Markov chain, a graphical model where inference (e.g., filtering, smoothing) has closed form solutions. We provide brief empirical results validating our theory.

## 1    INTRODUCTION

Probabilistic modeling of time-series data has applications ranging from robotic control (Theodorou et al., 2010) to material science (Jónsson et al., 1998) to cell biology (Saelens et al., 2019) to astrophysics (Majewski et al., 2017). These applications domains are often concerned with two questions: *predicting* future states (e.g., what will this cell look like in an hour); and *inferring* trajectories between two given states. However, answering these questions often require reasoning over high-dimensional data, which can be challenging as most tools in the standard probabilistic toolkit require generation.

One common approach is to learn compact representations of the data and then reason about those representations. The key challenge is to learn meaningful representations such that they continue to retain salient bits of information – simply optimizing for representations that are easy to predict results in degenerate representations (e.g., assign all observations a the zero-vector representation). For time-series data, we want the representation to remain a sufficient statistic for distributions related to time; for example, they should retain bits required to predict future states (or representations thereof). One approach that has this sufficiency property is reconstruction-based methods (Zhao et al., 2017; Zhu et al., 2020a; Makhzani et al., 2015; Dumoulin et al., 2016) – these methods aim to learn representations that contain all of the bits required to construct the high-dimensional observation $x$, so they must also contain the bits required to solve prediction and inference problems concerning $x$. However, reconstruction-based methods tend to be computationally expensive (see, e.g., (Razavi et al., 2019)) and can be challenging to scale to high-dimensional observations.

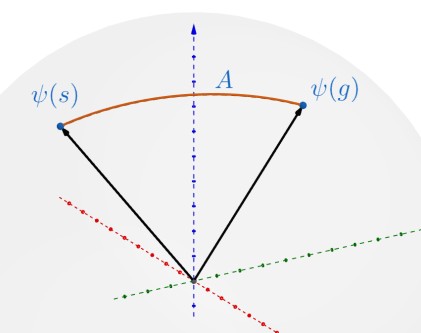

Figure 1: We will apply temporal contrastive learning to observations $(s, g)$ to obtain representations $(\psi(s), \psi(g))$ such that $A\psi(s)$ is close to $\psi(g)$. Using these representations, the distribution over intermediate representations has a closed form solution corresponding to linear interpolation between the initial and final representations.

In this paper, we start by observing that we want the learned representation to be a sufficient statistic for one observation's temporal relationship with other observations. In other words, our aim is to learn representations that contain information about observation $x$'s temporal relationships with other observations, but which need not retain other information in $x$ (e.g. pixel information). This intuition motivates us to study how contrastive representation learning methods (Oord et al., 2018; Sohn, 2016;

Chen et al., 2020b; Tian et al., 2020b; Wu et al., 2018) might be used to solve prediction and planning problems on time series data. While prior works in computer vision (Chen et al., 2020a; Oord et al., 2018) and natural language processing (NLP) (Mikolov et al., 2013) often study the geometry of learned representations, our results show why such interpolation is provably related to performing inference. Following prior work (Botvinick and Toussaint, 2012; Attias, 2003), we will use *planning* to refer to the problem of inferring intermediate states, not to refer to an optimal control problem.

The main contribution of this paper is showing that a certain form of temporal contrastive learning acquires representation wherein prediction and planning have closed form solutions. Specifically, we analyze symmetrized infoNCE objective (Radford et al., 2021), generating positive examples by sampling pairs of observations from the same time series data. Rather than constraining representations to have a fixed length, as done in prior work, we apply $L_2$ regularization to the representations. We provide analysis showing that the marginal distributions over learned representations should therefore be an isotropic Gaussian distribution, with variance corresponding to the average representation norm. By parametrizing the critic function as the negative MSE between representations of these observations, the learned probability ratio looks like a Gaussian potential function. With these pieces, we prove that the distribution over future representations has a Gaussian distribution, with a mean that is a linear function of the initial state representation. We extend this result to inference (i.e., planning): given an initial and final state, we show that the posterior distribution over an intermediate state's representation also follows a Gaussian distribution. We finally show that both of these results are special cases of a more general result: the distribution over representations behaves according to a Gaussian Markov chain, for which inference is easy (Malioutov et al., 2006; Weiss and Freeman, 1999) (See Fig. 1). In one special case, we show that intermediate representations have means evenly spaced on the line between the initial state representation and the final state representation.

## 2 RELATED WORK

**Representations for time-series data.** Time series representations are used in applications ranging from robotics to vision to NLP, where they enable users to reason about the temporal relationships between examples in terms of the spatial relationships of learned representations (Oord et al., 2018; Mikolov et al., 2013; Qian et al., 2021; Eysenbach et al., 2020). Ideally, these representations should retain information required to predict future observations and infer the likely plans (or trajectories) between pairs of observations. Many successful prior methods leverage reconstruction-based methods to learn such representations are able to reason about the sequential/temporal relationship of events along a trajectory to model similarity (Karamcheti et al., 2023; Park and Lee, 2021; Devlin et al., 2019; Carroll et al., 2022; Zhu et al., 2020b; Chung et al., 2015). While methods like a sequential VAE (Zhao et al., 2017) are transparent about how prediction and planning problems should be handled (as inference on the corresponding graphical model), they can be computationally expensive and challenging to scale to high-dimensional observations.

Contrastive learning methods circumvent reconstruction by learning representations that merely classify if two events were sampled from the same joint distribution (Gutmann and Hyvärinen, 2010; Chen and He, 2020; Radford et al., 2021). When applied to representing states along trajectories, contrastive representations learn to classify whether two points lie on the same trajectory or not (Oord et al., 2018; Sermanet et al., 2018; Eysenbach et al., 2022; Qian et al., 2021; Xu et al., 2023). Empirically, prior work in computer vision and NLP has observed that representations learned by various forms of contrastive learning acquires representations where interpolation between representations corresponds to changing the images in semantically meaningful ways (Wiskott and Sejnowski, 2002; Yan et al., 2021; Oring et al., 2020; Chen et al., 2019; Liu et al., 2018; Mikolov et al., 2013). Despite the computational benefits of contrastive representations, it is often unclear whether the learned representations are sufficient statistics for certain inference problems, and how such representations should be used for prediction and planning. The analysis in this paper shows how representations learned via temporal contrastive learning (i.e., without reconstruction) are sufficient statistics for inferring future outcomes, and can be reasoned about using the same language of graphical models typically associated with representations learned via reconstruction (e.g., the sequential VAE). Our analysis will provide a rigorous explanation for why we should expect representation interpolation to be meaningful for some contrastive methods.

Gaussian process methods (Roberts et al., 2013; Williams and Rasmussen, 2006) are an important class of models for time series data and have been successfully applied to control problems (Deisenroth and Rasmussen, 2011; Deisenroth et al., 2013). Unlike the Gaussian process, our method avoids

having to invert a matrix whose size depends on the number of data points. Additionally, while these prior methods typically assume that the input data are jointly Gaussian, we will look at learning compact *representations* that are jointly Gaussian.

**Goal-oriented decision making.** Many algorithms for goal-conditioned RL and similar settings employ spatiotemporal representations. This problem of goal-reaching dates to the early days of AI (Newell et al., 1959; Laird et al., 1987) but has received renewed attention in recent years (Chen et al., 2021; Chane-Sane et al., 2021; Colas et al., 2021; Yang et al., 2022; Ma et al., 2022b; Schroecker and Isbell, 2020; Janner et al., 2021). Some of the excitement in goal-conditioned RL is a reflection of the recent success of self-supervised methods in computer vision (e.g., Rombach et al. (2022)) and NLP (e.g., OpenAI (2023)). Our analysis will study a variant of contrastive representation learning proposed in prior work for goal-conditioned RL (Eysenbach et al., 2022; Sermanet et al., 2018). These methods are widespread, appearing as learning objectives for learning value functions (Eysenbach et al., 2020; 2022; Zheng et al., 2023; Tian et al., 2020a; Agarwal et al., 2019; Ma et al., 2022a; 2023; Nair et al., 2022; Liu et al., 2022; Wang et al., 2023), as auxiliary objectives (Schwarzer et al., 2020; Tang et al., 2022; Nair et al., 2022; Stooke et al., 2021; Bharadhwaj et al., 2022; Anand et al., 2019; Castro et al., 2021), in objectives for model-based RL (Shu et al., 2020; Ghugare et al., 2022; Allen, 2021; Mazoure et al., 2022), and in exploration methods (Guo et al., 2022; Du et al., 2021).

**Planning.** Planning lies at the core of many RL and control methods, allowing methods to infer the sequence of states and actions that would occur if the agent navigated from one state to a goal state (Attias, 2003; Thijssen and Kappen, 2015; Williams et al., 2015). These methods are often based on probabilistic inference, though the inference problem becomes challenging in high-dimensional settings. Some prior methods lift this limitation through non-parametric planning, inferring just a few intermediate states (Fang et al., 2022; Eysenbach et al., 2019; Zhang et al., 2021); however, it remains unclear how to scale these methods to high-dimensional settings when states do not lie on a low-dimensional manifold. Our analysis will show how planning in high-dimensional settings can be performed over (learned), low-dimensional representations, lifting the limitations of the aforementioned planning methods.

## 3 PRELIMINARIES

Broadly, our aim is to learn representations of time series data such that the spatial arrangement of representations corresponds to the temporal arrangement of the underlying data – if one example occurs shortly after another, then they should be mapped to similar representations. This problem setting arises in many areas, including video understanding and reinforcement learning. To define this problem formally, we will define a Markov process with states $x_t$ indexed by time $t$: $p(x_{1:T} \mid x_0) = \prod_{t=0}^{T} p(x_{t+1} \mid x_t)$. The standard RL setting can be subsumed into this by augmenting states with the previous action. The dynamics $p(x_{t+1} \mid x_t)$ tell us the immediate next state, and we can define the distribution over states $t$ steps in the future by marginalizing over the intermediate states, $p_t(x_t \mid x_0) = \int p(x_{1:t} \mid x_0)dx_{1:t-1}$. A key quantity of interest will be the $\gamma$-discounted state occupancy measure, which corresponds to a time-averaged distribution over future states:

$$p_{t+}(x_{t+} = x) = (1 - \gamma) \sum_{t=0}^{\infty} \gamma^t p_t(x_t = x). \tag{1}$$

**Contrastive learning.** Our analysis will focus on applying contrastive learning to a particular data distribution. Contrastive learning (Gutmann and Hyvärinen, 2010; Oord et al., 2018; Arora et al., 2019) acquires representations using "positive" pairs $(x, x^+)$ and "negative" pairs $(x, x^-)$. While contrastive learning typically learns just one representation, we will use two different representation for the two elements of the pair; that is, our analysis will use terms like $\phi(x)$, $\psi(x^+)$ and $\psi(x^-)$. We assume all representations lie in $\mathbb{R}^k$.

The aim of contrastive learning is to learn representations such that positive pairs have similar representations ($\phi(x) \approx \psi(x^+)$) while negative pairs have dissimilar representations ($\phi(x) \neq \psi(x^-)$). Let $p(x, x)$ be the joint distribution over positive pairs (i.e., $(x, x^+) \sim p(x, x)$); we will use the product of the marginal distributions to sample negative pairs ($(x, x^-) \sim p(x)p(x)$). Our analysis will use an infoNCE objective without resubstitution (Sohn, 2016; Oord et al., 2018):

$$\min_{\phi(\cdot), \psi(\cdot)} \quad \mathbb{E}_{\substack{(x,x^+) \sim p(x) \\ x_{1:B-1}^- \sim p(x)}} \left[ \log \frac{e^{f(\phi(x), \psi(x^+))}}{e^{\sum_{i=1}^{B-1} e^{f(\phi(x)^T, \psi(x_i^-))}}} \right]. \tag{2}$$

Contrastive learning is typically applied to an example $x$ and an augmentation $x^+$ of that same example; for example, in computer vision we might have image $x^+ \sim p(x \mid x)$ be a cropped version of image $x$. While prior work typically constrains the representations to have a constant norm (i.e., to lie on the unit hypersphere) (Oord et al., 2018), we will instead constrain the *expected* norm of the representations is bounded, a difference that will be important for our analysis:

$$\frac{1}{k} \mathbb{E}_{p(x)} \left[ \|\psi(x)\|_2^2 \right] \leq c. \tag{3}$$

Because the norm scales with the dimension of the representation, we have scaled down the left side by the representation dimension, $k$. In practice, we will impose this constraint by adding a regularization term $\lambda \mathbb{E}_{p(x)} \left[ \|\psi(x)\|_2^2 \right]$ to the infoNCE objective (Eq. 2) and dynamically tuning the weight $\lambda$ via dual gradient descent.

Because our aim is to learn representations that encode temporal information, we will follow prior work (Sermanet et al., 2018; Oord et al., 2018) in using some function of the *dynamics* to generate the positive pairs. While our experiments will sample positive examples from the discounted state occupancy measure ($x^+ \sim p_{t+}(x_{t+} \mid x)$) in line with prior work (Eysenbach et al., 2022), our analysis will also apply to different distributions (e.g., always sampling a state $k$ steps ahead).

### 3.1 KEY ASSUMPTIONS

This section outlines the two key assumptions behind our analysis.

**Assumption 1.** *the symmetrized infoNCE objective results in representations that encode a probability ratio, up to a constant $C$[1]:*

$$e^{-\frac{1}{2} \|\phi(x_0) - \psi(x)\|_2^2} = \frac{p_{t+}(x_{t+} = x \mid x_0)}{p(x)C}. \tag{4}$$

While prior work (Ma and Collins, 2018) has proven that the optimum of the contrastive learning objective satisfies this assumption, we nonetheless call this an "assumption" because this identity may not always hold in practice due to sampling and function approximation error. This identity means that the representations will encode the discounted state occupancy measure, up to a constant. Moreover, this assumption means the learned representations are sufficient statistics for predicting the probability (ratio) of future states. This property is essential, as it means that contrastive methods will not acquire degenerate representations (e.g., the zero-vector) that are easy to predict. Intuitively, these representations must retain all the information pertinent to reasoning about *temporal* relationships, but need not retain information about the precise contents of the observations. As such, they may be much more compressed than representations learned via reconstructive methods (e.g., a VAE).

Our analysis will also look at the marginal distribution over representations,

$$p(\psi) \triangleq \int p(x) \mathbb{1}(\psi(x) = \psi) dx.$$

**Assumption 2.** *Our second assumption, is that this marginal distribution is an isotropic Gaussian distribution:*

$$p(\psi) = \mathcal{N}(\psi; \mu = 0, \sigma = c \cdot I). \tag{5}$$

This assumption is important because it will allow us to express the distribution over sequences of representations as a Gaussian Markov chain. The denominator in Assumption 1 $p(x)$ may have a complex distribution, but Assumption 2 tells us that the distribution over *representations* has a simpler form. This will allow us to rearrange Assumption 1 to express the conditional distribution over representations as the product of two Gaussian likelihoods (note that the left hand side of Assumption 1 looks like a Gaussian likelihood).

In Appendix A.1 we provide some theoretical justification for why the marginal distribution over representations should be Gaussian; this is a minor extension of prior work (Wang and Isola, 2020). The key idea there (similar to (Shannon, 1948; Jaynes, 1957; Conrad, 2010)) is that maximum entropy distribution subject to an expected L2 norm constraint (Eq. 3) is an isotropic Gaussian distribution.

---

[1]While the result of Ma and Collins (2018) has $C(x)$ depending on $x$, the symmetrized version removes this dependence.

## 4 CONTRASTIVE REPRESENTATIONS MAKE PLANNING EASY

While inferring plans over arbitrary time series data is challenging, inference of Gaussian Markov chains is straightforward. In this section, we will show how representations learned by contrastive learning (with L2 regularization) are distributed according to a Gaussian Markov chain, making it straightforward to perform inference (e.g., planning, prediction) over these representations. Our proof technique will not contribute new analysis of Gaussian distributions, but rather combine (known) results about Gaussian distributions with (known) results about contrastive learning to produce a result that is not known (to the best of our knowledge): representations learned by temporal contrastive learning are distributed according to a Gaussian Markov chain. In the special case where dynamics are reversible (i.e., $p(x_t \mid x_0) = p(x_0 \mid x_t)$), planning will be equivalent to simple linear interpolation.

### 4.1 A PARAMETRIZATION FOR SHARED ENCODERS

This section describes the two encoders ($x \mapsto \psi$) we use to compute representations $\psi$ of $x$ and $x^+$. While prior work in computer vision and NLP literature use the same encoder for both $x$ and $x^+$, this decision does not make sense for many time-series data. Using the same encoder would imply that our prediction for $p(x_t \mid x_0)$ is the same as our prediction for $p(x_0 \mid x_t)$, which does not make sense for most Markov processes; the difficulty of transiting from $x_0$ to $x_t$ (e.g., climbing to the peak of a mountain) might be more difficult that the reverse (e.g., sledding down a mountain). At the same time, using entirely separate encoders is also insufficient for a subtle reason: it is unclear which encoder retains temporal information. It is plausible that $\phi(x_0)$ is a forward prediction of the future representation $\psi(x_t)$; it is equally plausible that $\psi(x_t)$ is a backward prediction of the previous representation $\phi(x_0)$. We will propose a parametrization that disambiguates which representation contains temporal information, which will be important for using these representations for planning.

We will treat the encoder $\psi(\cdot)$ as encoding the contents of the state. We will additionally learn a matrix $A$ so that the function $\psi \mapsto A\psi$ corresponds to a (multi-step) prediction of the future representation. To map this onto contrastive learning, we will use $\phi(x) \triangleq A\psi(x)$ as the encoder for the initial state. One way of interpreting this encoder is as an additional linear projection applied on top of $\psi(\cdot)$, a design similar to those used in other areas of contrastive learning (Chen and He, 2020). Once learned, we can use these encoders to answer questions about prediction (Sec. 4.2) and planning (Sec. 4.3).

### 4.2 REPRESENTATIONS ENCODE A PREDICTIVE MODEL

Given an initial state $x_0$, what states are likely to occur in the future? Answering this question directly in terms of high-dimensional states is challenging, but our learned representations provide a straightforward answer. Let $\psi_0 = \psi(x_0)$ and $\psi_{t+} = \psi(x_{t+})$ be random variables representing the representations of the initial state and a future state. Our aim is to estimate the distribution over these future representations, $p(\psi_{t+} \mid \psi_t)$. We will show that the learned representations encode this distribution.

**Lemma 4.1.** *Under the assumptions from Sec. 3, the distribution over representations of states from the discounted state occupancy measure follows a Gaussian distribution with mean parameter given by the initial state representation:*

$$p(\psi_{t+} = \psi \mid \psi_0) = \mathcal{N}\left(\mu = \frac{c}{c+1}A\psi_0, \Sigma = \frac{c}{c+1}I\right). \tag{6}$$

The main takeaway here is that the distribution over future representations has a convenient, closed form solution. The representation norm constraint, $c$, determines the shrinkage factor $\frac{c}{c+1} \in [0, 1)$; highly regularized settings (small $c$) move the mean closer towards the origin and decrease the variance. Regardless of the constraint $c$, the predicted mean is a linear function $\psi \mapsto \frac{c}{c+1}A\psi$.

*Proof.* Our proof technique will be similar to that of the law of the unconscious statistician:

$$p(\psi_{t+} \mid \psi_0) \overset{(a)}{=} \frac{p(\psi_{t+} \mid \psi_0)}{p(\psi_0)}$$

$$\propto \iint p(\psi_{t+}, x_{t+}, \psi_0, x_0) dx_{t+} dx_0$$

$$\overset{(b)}{=} \iint p(\psi_{t+} \mid x_{t+}) p(\psi_0 \mid x_0) p(x_{t+} \mid x_0) p(x_0) dx_{t+} dx_0$$

$$\overset{(c)}{\propto} \iint \mathbb{1}(\psi(x_{t+}) = \psi_{t+})\mathbb{1}(\psi(x_0) = \psi_0)p(x_{t+})e^{-\frac{1}{2}\|A\psi(x_0)-\psi(x)\|_2^2}p(x_0)dx_{t+}dx_0$$

$$\overset{(d)}{=} e^{-\frac{1}{2}\|A\psi_0-\psi_{t+}\|_2^2} \iint \mathbb{1}(\psi(x_{t+}) = \psi_{t+})\mathbb{1}(\psi(x_0) = \psi_0)p(x_{t+})p(x_0)dx_{t+}dx$$

$$\overset{(e)}{=} e^{-\frac{1}{2}\|A\psi_0-\psi_{t+}\|_2^2} \underbrace{\left(\int p(x_{t+})\mathbb{1}(\psi(x_{t+}))dx_{t+}\right)}_{p(\psi_{t+})} \underbrace{\left(\int p(x_0)\mathbb{1}(\psi(x_0)dx_0\right)}_{p(\psi_0)}$$

$$\overset{(f)}{\propto} e^{-\frac{1}{2}\|A\psi_0-\psi_{t+}\|_2^2}e^{-\frac{1}{2c}\|\psi_{t+}\|_2^2}e^{-\frac{1}{2c}\|\psi_0\|_2^2}$$

$$\overset{(g)}{\propto} e^{-\frac{1+\frac{1}{c}}{2}\|\frac{1}{1+\frac{1}{c}}A\psi_0-\psi_{t+}\|_2^2} \propto \mathcal{N}\left(\psi_{t+}; \mu = \tfrac{c}{c+1}A\psi_0, \Sigma = \tfrac{c}{c+1}I\right).$$

In *(a)* we applied Bayes' Rule and remove the denominator, which is a constant w.r.t. $\psi_{t+}$. In *(b)* we factored the joint distribution, noting that $\psi_{t+}$ and $\psi_0$ are deterministic functions of $x_{t+}$ and $x_0$ respectively, so they are conditionally independent from the other random variables. In *(c)* we used Assumption 1 after solving for $p(x_{t+} \mid x_0) = p(x_{t+})e^{-\frac{1}{2}\|A\psi(x_0)-\psi(x)\|_2^2}$. In *(d)* we noted that when the integrand is nonzero, it takes on a constant value of $e^{-\frac{1}{2}\|A\psi_0-\psi_{t+}\|_2^2}$, so we can move that constant outside the integral. In *(e)* we use the definition of the marginal representation distribution (Eq. 6). In *(f)* we use Assumption 2 to write the marginal distributions $p(\psi_{t+}), p(\psi_0)$ as Gaussian distributions; we removed the normalizing constants, which are independent of $\psi_{t+}$. In *(h)* we completed the square and then recognized the expression as the density of a multivariate Gaussian distribution. □

### 4.3 Planning over One Intermediate State

We now show how these representations can be used for planning. We refer to one specific type of planning problem: given an initial state $x_0$ and a future state $x_{t+}$, our aim will be to infer (the representation of) an intermediate "waypoint" state $x_w$. Our aim is not to infer the entire sequence of intermediate states. We assume $x_0 \to x_w \to x_{t+}$ form a Markov chain where $x_w \sim p(x_{t+} \mid x_0 = x_0)$ and $x_{t+} \sim p(x_{t+} \mid x_0 = x_w)$ are both drawn from the discounted state occupancy measure (Eq. 1). Let random variable $\psi_w = \psi(x_w)$ be the representation of this intermediate state. Our main result is that the posterior distribution over waypoint *representation* has a closed form solution in terms of the initial state representation and future state representation:

**Lemma 4.2.** *Under Assumptions 1 and 2, the posterior distribution over waypoint representations is a Gaussian whose mean and covariance are linear functions of the state and goal representations:*

$$p(\psi_w \mid \psi_0, \psi_{t+}) = \mathcal{N}\left(\psi_w; \mu = \Sigma^{-1}(A^T\psi_{t+} + A\psi_0), \Sigma = \tfrac{c}{c+1}A^TA + \tfrac{c+1}{c}I\right)$$

The proof (Appendix A.2) uses the Markov property together with Lemma 4.1. The main takeaway from this lemma is that the posterior distribution takes the form of a simple probability distribution (a Gaussian) with parameters that are linear functions of the initial and final representations.

We give three examples to build intuition:

**Example 1:** $A = I$ and the $c$ is very large (little regularization). Then, the covariance is $\Sigma \approx 2I$ and the mean is the simple average of the initial and final representations $\mu \approx \frac{1}{2}(\psi_0 + \psi_{t+})$. In other words, the waypoint representation is the midpoint of the line $\psi_0 \to \psi_{t+}$.

**Example 2:** $A$ is a rotation matrix and $c$ is very large. Rotation matrices satisfy $A^T = A^{-1}$ so the covariance is again $\Sigma \approx 2I$. As noted in Sec. 4.2, we can interpret $A\psi_0$ as a *prediction* of which representations will occur after $\psi_0$. Similarly, $A^{-1}\psi_{t+} = A^T\psi_{t+}$ is a prediction of which representations will occur before $\psi_{t+}$. Lemma 4.2 tells us that the mean of the waypoint distribution is the simple average of these two predictions, $\mu \approx \frac{1}{2}(A^T\psi_{t+} + A\psi_0)$.

**Example 3:** $A$ is a rotation matrix and $c = 0.01$ (very strong regularization). In this case $\Sigma = \frac{0.01}{0.01+1}A^TA + \frac{0.01+1}{0.01}I \approx 100I$, so $\mu \approx \frac{1}{100}(\psi_0 + \psi_{t+}) \approx 0$. Thus, in the case of strong regularization the posterior is highly concentrated around the origin.

### 4.4 PLANNING OVER MANY INTERMEDIATE STATES

This section extends the previous section to consider multiple intermediate states. Again, we will infer the posterior distribution of the representations of these intermediate states, $\psi_{w_1}, \psi_{w_2}, \cdots$. As before, we will assume that these states form a Markov chain where the conditional distribution is given by the discounted state occupancy measure. For example, in the case of two intermediate states, we can write the posterior as:

$$p(\psi_{w_1}, \psi_{w_2} \mid \psi_0, \psi_{t+}) = \frac{p(\psi_{t+}|\psi_{w_2})p(\psi_{w_2}|\psi_{w_1})p(\psi_{w_1}|\psi_0)}{p(\psi_{t+}|\psi_0)}$$

$$\propto e^{-\frac{1+\frac{1}{c}}{2}\|\frac{c}{c+1}A\psi_{w_2}-\psi_{t+}\|_2^2 - \frac{1+\frac{1}{c}}{2}\|\frac{c}{c+1}A\psi_{w_1}-\psi_{w_2}\|_2^2 - \frac{1+\frac{1}{c}}{2}\|\frac{c}{c+1}A\psi_0-\psi_{w_1}\|_2^2}.$$

In the general case of $n$ intermediate states, the posterior distribution is

$$p(\psi_{w_1}, \cdots, \psi_{w_n} \mid \psi_0, \psi_{t+}) \propto e^{-\frac{1+\frac{1}{c}}{2}\sum_{i=1}^{n}\|\frac{c}{c+1}A\psi_{w_i}-\psi_{w_{i+1}}\|_2^2},$$

where $\psi_{w_0} = \psi_0$ and $\psi_{w_{n+1}} = \psi_{t+}$. This corresponds to a chain graphical model with edge potentials $f(\psi, \psi') = e^{-\frac{1+\frac{1}{c}}{2}\|\frac{c}{c+1}A\psi-\psi'\|_2^2}$. This posterior distribution corresponds to a joint Gaussian distribution over a long vector of concatenated representations $\psi_{1:n} = (\psi_{w_1}, \cdots, \psi_{w_n})$:

$$p(\psi_{1:n}) \propto \exp\left(-\tfrac{1}{2}\psi_{1:n}^T \Sigma^{-1}\psi_{1:n} + \eta^T\psi_{1:n}\right),$$

where $\Sigma$ is a tridiagonal matrix

$$\Sigma^{-1} = \begin{pmatrix} \frac{c}{c+1}A^TA+\frac{c+1}{c}I & -A^T & \\ -A & \frac{c}{c+1}A^TA+\frac{c+1}{c}I & -A^T \\ & & \ddots \end{pmatrix}, \quad \text{and } \eta = \begin{pmatrix} A\psi_0 \\ 0 \\ \vdots \\ \vdots \\ A^T\psi_{t+} \end{pmatrix}.$$

This distribution can be written in the canonical parametrization as $\Sigma = \Lambda^{-1}$ and $\mu = \Sigma\eta$. Recall that Gaussian distributions are closed under marginalization. Thus, once in this canonical parametrization, the marginal distributions can be obtained by reading off individual entries of these parameters:

$$p(\psi_i \mid \psi_0, \psi_{t+}) = \mathcal{N}\left(\psi_i; \mu_i = (\Sigma\eta)^{(i)}, \Sigma_i = (\Lambda^{-1})^{(i,i)}\right).$$

The key takeaway here is that this posterior distribution over waypoints is Gaussian, and it has a closed form expression in terms of the initial and final representations (as well as the learned matrix $A$ and regularization parameter $c$).

**Special case.** To build intuition, consider the special case where $A$ is a rotation matrix and $c$ is very large, so $\frac{c}{c+1}A^TA + \frac{c+1}{c} \approx 2I$. In this case, $\Sigma^{-1}$ is a (block) second difference matrix (Higham, 2022):

$$\Sigma^{-1} = \begin{pmatrix} 2I & -I & \\ -I & 2I & -I \\ & & \ddots \end{pmatrix}.$$

The inverse of this matrix has a closed form solution (Newman and Todd, 1958, Pg. 471), allowing us to obtain the mean of each waypoint in closed form:

$$\mu_i = (1 - \lambda(i))\psi_0 + \lambda(i)A^T\psi_{t+}, \quad \text{where } \lambda(i) = \frac{i}{n+1}.$$

Thus, each posterior mean is a convex combination of the (forward prediction from the) initial representation and the (backwards prediction from the) final representation. When $A$ is the identity matrix, the posterior mean is simple linear interpolation between the initial and final representations!

## 5 NUMERICAL SIMULATION

While the contributions of this paper are theoretical, we include several didactic experiments to illustrate our results. Code to reproduce these results, including all hyperparameters, is included in the Supplemental Materials.

### 5.1 SYNTHETIC DATASET

Fig. 2 (top left) shows a dataset of time series data, starting at the origin and spiraling outwards. We applied contrastive learning with the parametrization in Sec. 4.1 to these data. We then used these representations to solve prediction and planning problems using the derivations in Sec. 4.2 and Sec. 4.3.

DATASET  FORWARD PREDICTION

■ start ($x_0$)
★ end ($x_T$)

BACKWARD PREDICTION  PLANNING

Figure 2: **Numerical simulation of our analysis.** *(Top Left)* Toy dataset of time-series data consisting of many outwardly-spiraling trajectories. We apply temporal contrastive learning to these data. *(Top Right)* For three initial observations (■), we use the learned representations to predict the distribution over future observations. Note that these distributions correctly capture the spiral structure. *(Bottom Left)* For three initial observations (★), we use the learned representations to predict the distribution over preceding observations. *(Bottom Right)* We plot the distribution over one waypoint (Sec. 4.3). The representations capture the shape of the distribution, though they do mistakenly assign non-zero probability mass to parts of the spiral inside of ■ and outside of ★.

Fig. 2 (top right) shows forward predictions, while Fig. 2 (bottom left) shows backwards predictions. Note that these predictions correctly handle the nonlinear structure of these data – states nearby the initial state in Euclidean space that are not temporally adjacent are assigned low likelihood. Fig. 2 (bottom right) shows the posterior distribution over one waypoint (planning).

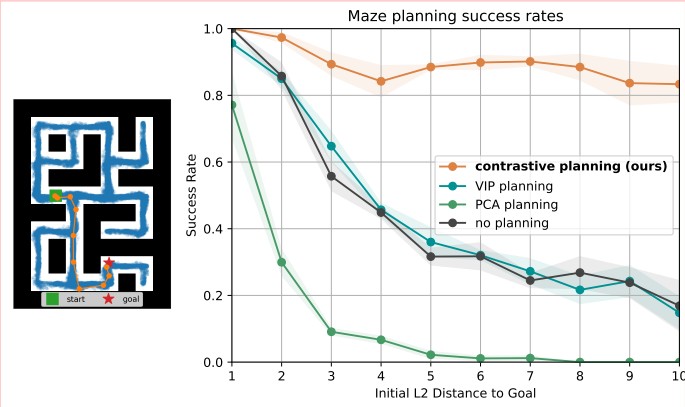

Figure 3: Using inferred paths over our contrastive representations for control boosts success rates by $4.5\times$ on the most difficult goals ($18\% \rightarrow 84\%$). Baseline representation learning techniques fail to improve performance when used for planning.

## 5.2 SOLVING MAZES WITH INFERRED REPRESENTATIONS

Our next experiment studies whether the inferred representations are useful for solving a control task. We took a 2d maze environment and dataset from prior work (Fig. 3, left) (Fu et al., 2020) and learned encoders from this dataset. To solve the maze, we take the observation of the starting state and goal state, compute the representations of these states, and use the analysis in Sec. 4.3 to infer the sequence of intermediate representations. We visualize the results using a nearest neighbor retrieval (Fig. 3 left). Appendix Fig. 5 contains additional examples.

Finally, we studied whether these representations are useful for control. We implemented a simple proportional controller for this maze. As expected, this proportional controller can successfully navigate to close goals, but fails to reach distant goals (Fig. 3, right). However, if we use the proportional controller to track a series of waypoints planned using our representations (i.e., the orange dots shown in Fig. 3 (left)), the success rate increases by up to $4.5\times$. Unlike our contrastive representations, baseline representation approaches like PCA and VIP (Ma et al., 2022a) fail to generate waypoints for planning that improve performance.

## 5.3 HIGHER DIMENSIONAL TASKS

In this section we provide preliminary experiments showing the planning approach in Sec. 4 scales to higher dimensional tasks. We used two datasets from prior work (Fu et al., 2020): `door-human-v0` (39-dimensional observations) and `hammer-human-v0` (46-dimensional observations). After learning encoders on these tasks, we evaluated the inference capabilities of the learned representations. We sampled (unseen) trajectories and compared the inferred sequence of planned waypoints (using a nearest neighbor) to the ground truth sequence of states. As shown in Fig. 4 and Appendix Fig. 6, the

inferred representations roughly align with the ground truth plan, seeming to do so more accurately than representations from PCA and VIP (with simple linear interpolation). Further, Fig. 4 shows quantitative comparisons of the error of predicted waypoint plans across our approach, PCA and VIP representations, and a naive no-planning baseline. According to the MSE error metric, our approach produces significantly better plans than all the baselines (Fig. 4, top right).

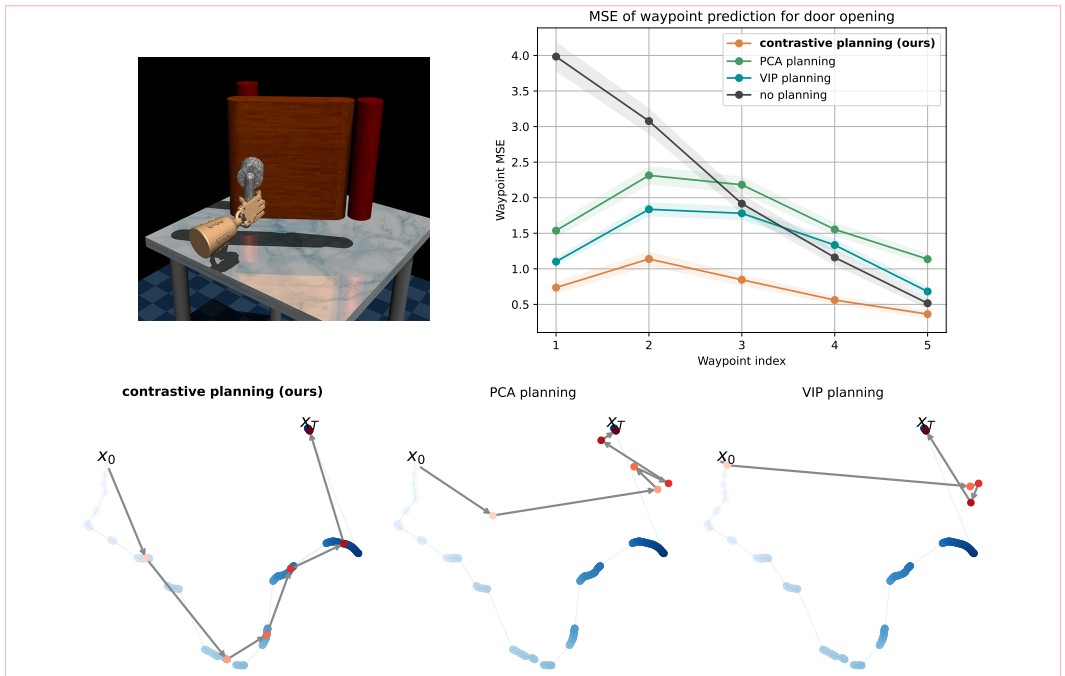

Figure 4: Planning for 39-dimensional robotic door opening. *(Top Left)* We use a dataset of trajectories demonstrating door opening from prior work (Fu et al., 2020) to learn representations. *(Top Right)* We compute the MSE for the five planned waypoints produced by our method against the true observations at the corresponding points in the trajectory in 39-dimensional space (error bars $\pm$ 1 standard error). We compare against baseline plans generated by interpolating PCA and VIP representations, as well as a "no planning" baseline which simply always predicts the goal of the trajectory as each waypoint. *(Bottom)* We visualize a TSNE of the states along sample trajectory as blue circles, with the transparency indicating the index along the trajectory. *(Bottom Left)* We visualize the inferred plan (Sec. 4.3) as red circles connected by arrows. *(Bottom Right)* We also visualize the plans generated by the PCA and VIP representations in the TSNE space. Our contrastive representations plan waypoints that better fit the intermediate states of trajectory.

## 6 DISCUSSION

Representation learning is at the core of many high-dimensional time-series modeling questions, yet how those representations are learned is often disconnected with the inferential task. The main contribution of this paper is to show how *discriminative* techniques can be used to acquire compact representations that make it easy to answer inferential questions about time. The precise objective and parametrization we studied is not much different from that used in practice, suggesting that either our theoretical results might be adapted to the existing methods, or that practitioners might adopt these details so they can use the closed-form solutions to inference questions. Our work may also have implications for studying the structure of learned representations. While prior work often studies the geometry of representations as a post-hoc check, our analysis may provide tools for studying *when* interpolation properties are guaranteed to emerge, as well as *how* to learn representations with certain desired geometric properties.

**Limitations.** Our analysis hinges on the two assumptions mentioned in Sec. 3.1, and it remains open how errors in those approximations translate into errors in our analysis. Structurally, this paper aims to provide some mathematical tools for studying these contrastive representations, but applying these techniques in real-world application domains remains an important problem for future work.

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

# A   PROOFS

## A.1   MARGINAL DISTRIBUTION OVER REPRESENTATIONS IS GAUSSIAN

The infoNCE objective (Eq. 2) can be decomposed into an alignment term and a uniformity term (Wang and Isola, 2020), where the uniformity term can be simplified as follows:

$$
\mathbb{E}_{x \sim p(x)} \left[ \log \mathbb{E}_{x^- \sim p(x)} \left[ e^{-\frac{1}{2} \| A\psi(x^-) - \psi(x) \|_2^2} \right] \right]
$$

$$
= \frac{1}{N} \sum_{i=1}^{N} \log \left( \frac{1}{N-1} \sum_{j=1\cdots N, j \neq i} e^{-\frac{1}{2} \| A\psi(x_i) - \psi(x_j) \|_2^2} \right)
$$

$$
= \frac{1}{N} \sum_{i=1}^{N} \log \left( \frac{1}{N-1} \sum_{j=1\cdots N, j \neq i} \underbrace{\frac{1}{(2\pi)^{k/2}} e^{-\frac{1}{2} \| A\psi(x_i) - \psi(x_j) \|_2^2}}_{\mathcal{N}(\mu = \psi(x_j); \Sigma = I)} \right) + \frac{k}{2} \log(2\pi)
$$

$$
= \frac{1}{N} \sum_{i=1}^{N} \log \hat{p}_{\text{GMM}}(\psi(x_i)) + \frac{k}{2} \log(2\pi)
$$

$$
= -\hat{\mathcal{H}}[\psi(x)] + \frac{k}{2} \log(2\pi).
$$

The derivation above extends that in Wang and Isola (2020) by considering a Gaussian distribution rather than a von Mises Fisher distribution. We are implicitly making the assumption that the marginal distributions satisfy $p(x) = p(x^-)$. This difference corresponds to our choice of using a negative squared L2 distance in the infoNCE loss rather than an inner product, a difference that will be important later in our analysis. A second difference is that we do not use the resubstitution estimator (i.e., we exclude data point $x_i$ from our estimate of $\hat{p}_{\text{GMM}}$ when evaluating the likelihood of $x_i$), which we found hurt performance empirically. The takeaway from this identity is that maximizing the uniformity term corresponds to maximizing (an estimate of) the entropy of the representations.

We next prove that the maximum entropy distribution with an expected L2 norm constraint is a Gaussian distribution. Variants of this result are well known (Shannon, 1948; Jaynes, 1957; Conrad, 2010), but we include a full proof here for transparency.

**Lemma A.1.** *The maximum entropy distribution satisfying the expected L2 norm constraint in Eq. 3 is a multivariate Gaussian distribution with mean $\mu = 0$ and covariance $\Sigma = c \cdot I$*

*Proof.* We start by defining the corresponding Lagrangian, with the second constraint saying that $p(x)$ must be a valid probability distribution.

$$
\mathcal{L}(p) = \mathcal{H}_p[x] + \lambda_1 \left( \mathbb{E}_{p(x)} \left[ \| x \|_2^2 \right] - c \cdot k \right) + \lambda_2 \left( \int p(x) dx - 1 \right)
$$

We next take the derivative w.r.t. $p(x)$:

$$
\frac{\partial \mathcal{L}}{\partial p(x)} = -p(x)/p(x) - \log p(x) + \lambda_1 \| x \|_2^2 + \lambda_2
$$

Setting this derivative equal to 0 and solving for $p(x)$, we get

$$
p(x) = e^{-1 + \lambda_2 + \lambda_1 \| x \|_2^2}.
$$

We next solve for $\lambda_1$ and $\lambda_2$ to satisfy the constraints in the Lagrangian. Note that $x \sim \mathcal{N}(\mu = 0, \Sigma = c \cdot I)$ has an expected norm $\mathbb{E}[\| x \|_2^2] = c \cdot k$, so we must have $\lambda_1 = -\frac{1}{2c}$. We determine $\lambda_1$ as the normalizing constant for a Gaussian, finally giving us:

$$
p(x) = \frac{1}{(2c\pi)^{k/2}} e^{\frac{-1}{2c} \| x \|_2^2}
$$

corresponding to an isotropic Gaussian distribution with mean $\mu = 0$ and covariance $\Sigma = c \cdot I$.   □

## A.2 PROOF OF LEMMA 4.1: WAYPOINT DISTRIBUTION

*Proof.*

$$p(\psi_w \mid \psi_0, \psi_{t+}) \overset{(a)}{=} \frac{p(\psi_{t+} \mid \psi_w)p(\psi_w \mid \psi_0)}{p(\psi_{t+} \mid \psi_0)}$$

$$\overset{(b)}{\propto} e^{-\frac{1+\frac{1}{c}}{2}\|\frac{c}{c+1}A\psi_w - \psi_{t+}\|_2^2} e^{-\frac{1+\frac{1}{c}}{2}\|\frac{c}{c+1}A\psi_0 - \psi_w\|_2^2}$$

$$\overset{(c)}{\propto} e^{-\frac{1}{2}(\psi_w - \mu)^T \Sigma (\psi_w - \mu)} \quad \text{where } \Sigma = \frac{c}{c+1}A^T A + \frac{c+1}{c}I, \ \mu = \Sigma^{-1}(A^T\psi_{t+} + A\psi_0).$$

$$= \mathcal{N}(\psi_w; \mu, \Sigma).$$

$\square$

In line *(a)* we used the definition of the conditional distribution and then simplified the numerator using the Markov property. Line *(b)* uses the Lemma 4.1. Line *(c)* completes the square, using $\overset{\text{const.}}{=}$ to denote equality up to an additive constant that is independent of $\psi_w$, and using the definitions of $\mu$ and $\Sigma$ above:

$$\frac{c+1}{c}\left(\left\|\frac{c}{c+1}A\psi_w - \psi_{t+}\right\|_2^2 + \left\|\frac{c}{c+1}A\psi_0 - \psi_w\right\|_2^2\right)$$

$$= \frac{c+1}{c}\left(\psi_w^T(\frac{c}{c+1}A)^T(\frac{c}{c+1}A)\psi_w - 2\psi_{t+}^T(\frac{c}{c+1}A)\psi_w + \psi_{t+}^T\psi_{t+}\right.$$

$$\left. + \psi_0^T(\frac{c}{c+1}A)^T(\frac{c}{c+1}A)\psi_0 - 2\psi_0^T(\frac{c}{c+1}A)^T\psi_w + \psi_w^T\psi_w\right)$$

$$\overset{\text{const.}}{=} \psi_w^T\underbrace{(\frac{c}{c+1}A^T A + \frac{c+1}{c}I)}_{\Sigma}\psi_w - 2(A^T\psi_{t+} + A\psi_0)^T\psi_w$$

$$\overset{\text{const.}}{=} (\psi_w - \mu)^T \Sigma(\psi_w - \mu),$$

where $\Sigma = \frac{c}{c+1}A^T A + \frac{c+1}{c}I$ and $\mu = \Sigma^{-1}(A^T\psi_{t+} + A\psi_0)$.

## B   ADDITIONAL EXPERIMENTS

Fig. 5 visualizes the inferred waypoints from the task in Fig. 3.

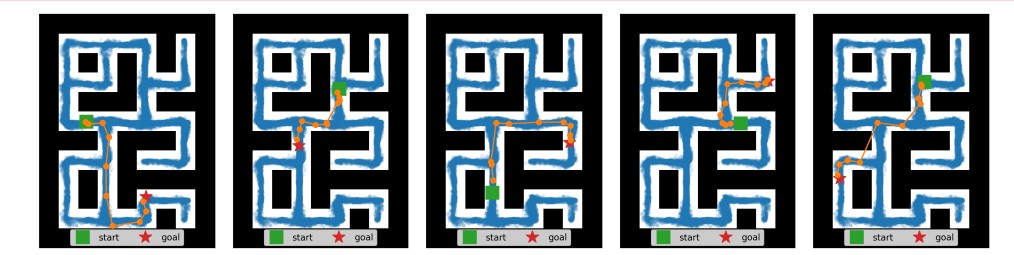

Figure 5: Our approach enables a goal-conditioned policy to reach farther targets (red) from the start (green) by planning over intermediate waypoints (orange).

Fig. 6 visualizes the representations learned on a 46-dimensional robotic hammering task (see Sec. 5.3).

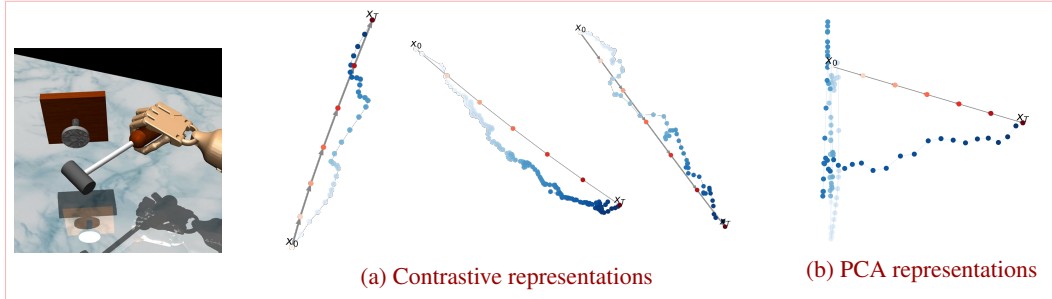

(a) Contrastive representations            (b) PCA representations

Figure 6: Planning for 46-dimensional robotic hammering. *(Left)* A dataset of trajectories demonstrating a hammer knocking a nail into a board (Fu et al., 2020). *(Center)* We visualize the learned representations as blue circles, with the transparency indicating the index of that observation along the trajectory. We also visualize the inferred plan (Sec. 4.3) as red circles connected by arrows. *(Right)* Representations learned by PCA on the same trajectory as *(a, left)*.

