# OpenReview forum: "Contrastive Representations Make Planning Easy"
_ICLR.cc/2024/Conference — Submitted to ICLR 2024_

### Official Review · Reviewer_RwSP · 2023-10-23

**Soundness:** 2 fair
**Presentation:** 2 fair
**Contribution:** 3 good
**Rating:** 3
**Confidence:** 3

**Summary:**

This blue sky research paper presents a contrastive learning method for time series. Its key selling point is that temporal dynamics have closed form solutions. Once the heavy assumptions on everything being Gaussian are digested, well known properties of Gaussians (like marginal distributions again being Gaussian) can be exploited.

**Strengths:**

The ability to inter- and extrapolate trajectories in closed form is surely a great and extremely powerful property.

Full code is included, even if the experiment is minimal at best.

**Weaknesses:**

The by far weakest point of the paper is that the authors miss to demonstrate that their method works on a real problem. After all, this is still machine learning. No matter how beautiful the derivation is, a method that does not work in the real world has no value. I have severe doubts that it will every work, say, for modeling an industrial process or a robot turning a knob to open a door. For my taste, the assumptions in section 3 are far too strong to be left unchecked on real data.

The paper has multiple presentation and language issues:
1. Use of terms: The term "planning" has a technical meaning in artificial intelligence. That meaning is very different from temporal inter- or extrapolation, which is what it is refers to in the paper. Therefore, even the paper title is completely misleading. A similar consideration applies to the central term "representation", which is used as a map from observations to a latent space, and also for an element of that space, sometimes with different meaning within the same sentence.
2. Abbreviations: The abbreviations "CV" and "NLP" are of course generally understood in the community. It is nevertheless good style to state the complete terms at least once before using abbreviations. Other abbreviations are less commonly known, like NCE.
3. Figure 1 is pretty meaningless.
4. The first paragraph of section 2 ends in the middle of a sentence.
5. I am surprised to see that the title of section 4 coincides with the paper title.
6. At the point figure 2 is presented (section 1), the symbols in the figure are not introduced yet.
7. Why on earth should one gray out a proof?

**Questions:**

I don't have any questions that need clarification in the rebuttal phase.

---

> ### Author Response · Authors · 2023-11-17
> **Author response: new experiments and revisions**
>
> Dear Reviewer,
>
> We thank the reviewer for the detailed feedback. It seems like the main suggestion is to add more experiments, which we have done through new experiments on higher-dimensional tasks (up to 46-dimensional) and by demonstrating that the inferred representations are useful for control. **Together with the revisions (red text in new PDF) and clarifications discussed below, does this fully address the reviewer's concerns about the paper?** We look forward to continuing the discussion!
>
> > demonstrate that their method works on a real problem … robot turning a knob to open a door
>
> As suggested, we have run additional experiments to validate our theory, including on a 39-dim robot door opening task:
> * [Fig 3 (left)](https://i.imgur.com/BMAg3yU.png): We show how the analysis in Sec. 4 can be used to solve a maze. After learning representations, we used the closed form expression to infer the representations of intermediate states. This result shows how the representations effectively warp the state space so that linear motion in representations (i.e., $A \psi$) corresponds to fairly complex motion in the state space (i.e., navigating through a maze).
> * [Fig 3 (right)](https://i.imgur.com/BMAg3yU.png): Using this same maze, we show that the inferred waypoints can be used for control (details below).
> * [Fig 4](https://i.imgur.com/XgDrGOu.png): We apply the representations to a 39-dimensional robotic door opening task. Compared to a PCA baseline, linearly interpolated waypoints are a better fit for the true data waypoints.
> * [Fig 6](https://i.imgur.com/gVnReJj.png): We apply the representations to a 46-dimensional robotic hammering task. Compared to a PCA baseline, linearly interpolated waypoints are a better fit for the true data waypoints.
>
> > The term "planning" has a technical meaning in artificial intelligence. That meaning is very different from temporal inter- or extrapolation, which is what it refers to in the paper.
>
> Thanks for pointing this out! We agree that there is no single clear meaning of this word and it is often misused. We have revised the introduction to explain that we will use ``planning’’ to refer to an inference problem, rather than an optimal control problem. We provide citations there to some prior work that takes a similar perspective (Botvinick 2012, Attias 2003), though we acknowledge that this may be a nonstandard usage within the AI community.
>
> > "representation", which is used as a map from observations to a latent space, and also for an element of that space
>
> Thanks for raising this great point! It indeed seems like a potential source of confusion. To incorporate this feedback, we have significantly revised Sec. 4.1 to disambiguate “encoder” from “representation.” We have also reviewed every use of the word “representation” in the rest of the main text, making some minor changes in attempts to clarify this. **Are there remaining parts of the paper where this is unclear?** We would be happy to make additional changes to improve clarity and precision.
>
> > Abbreviations: CV, NLP
>
> We have written these terms out in full the first time they are used.
>
> > Figure 1 is pretty meaningless.
>
> As suggested, we have removed Figure 1.
>
> > The first paragraph of section 2 ends in the middle of a sentence.
>
> We have completed the sentence in the revision.
>
> > I am surprised to see that the title of section 4 coincides with the paper title.
>
> Our intention here was to make it easy for readers to identify where to find the main results. We also felt like it was an accurate description of the section. If the current title is unclear or imprecise, we would be happy to revise the section title.
>
> > At the point figure 2 is presented (section 1), the symbols in the figure are not introduced yet.
>
> We have revised the caption to briefly explain the symbols
>
> > Why on earth should one gray out a proof?
>
> We have removed this. Our original intention was to indicate that readers could skip over this section if they were not interested in the proof, but we realize in hindsight it can cause confusion.

---

> > ### Author Response · Authors · 2023-11-20
> > **Do the new experiments (e.g., robotic door opening) and revisions address the concerns?**
> >
> > Dear Reviewer,
> >
> > Do the new experiments (including robotic door opening) and revisions described above fully address the concerns about the paper? We believe that these changes, motivated by the great feedback, strengthens the paper. While the time remaining in the review period is limited, we would be happy to try to run additional experiments or make additional revisions.
> >
> > Kind regards,
> >
> > The Authors

---

### Official Review · Reviewer_QatW · 2023-10-31

**Soundness:** 3 good
**Presentation:** 3 good
**Contribution:** 1 poor
**Rating:** 5
**Confidence:** 3

**Summary:**

The paper aims at leveraging contrastive representation for planning. It assumes representations in a sequential decision problems, say RL setting, are learned by contrastive learning, and under certain assumptions, the authors derive the conditional probability distribution of future state representation given initial state, and that of intermediate states’ representation given an initial and an ending representations. Simple empirical results are provided to validate the proposed method.

**Strengths:**

1. The topic of planning in representation space looks interesting;

2. I am not aware of works studying contrastive learning for representation learning and planning, it appears to be novel in this regard

**Weaknesses:**

My primary concern is in the significance of this work.

The presented result, lemma 4.1, 4.2 seem to follow a commonly seen algebraic computation when deriving multivariate conditional gaussian distributions, there is no new contribution in terms of proof techniques.

Though planning is important in control, this paper focuses on the calculation of the conditional probability given certain states’ representations, I do not see why this is a difficult task. It is often more critical to show how effective the sampled states can be used in planning, which is not studied in this paper.

The main theoretical results lemma 4.1, 4.2 alone seem not enough for a top conference. The authors should discuss the connection to several related work: PILCO work by Marc et al., and Gaussian processes for data-efficient learning in robotics and control. Although the proposed methods are not the same, the involved computation bears similarities and all works involves planning. The authors might agree that the two works contain more contributions both theoretically and empirically.

**Questions:**

see above.

---

> ### Author Response · Authors · 2023-11-17
> **Author response: significant revisions and new experiments**
>
> Dear Reviewer,
>
> We thank the reviewer for the detailed feedback. It seems like the reviewer's main concern is about the significance of the work. We have revised the paper to clarify the significance (start of Sec. 4), and provide more details below. As suggested, we have also added a new experiment to demonstrate how the sampling states (based on our representations) can be used for planning ([Fig 4](https://i.imgur.com/XgDrGOu.png)). **Do these revisions address the reviewer's concerns about significance?** We look forward to continuing the discussion.
>
> > how effective the sampled states can be used in planning, which is not studied in this paper.
>
> We ran an additional experiment to study how the learned representations might be used for control ([new Fig. 3](https://i.imgur.com/BMAg3yU.png)). These results show that the inferred representations can boost the performance of a simple proportional controller by up to 4.5x (for reaching the most distant goals).
>
> > there is no new contribution in terms of proof techniques.
>
> We agree that our paper does not contribute new analysis of Gaussian distributions; rather, it combines (known) results about Gaussian distributions with (known) results about contrastive learning (that it learns a probability ratio) to produce a result that is not known (to the best of our knowledge): that representations learned by temporal contrastive learning are distributed according to a Gaussian Markov chain. We have revised the start of Sec. 4 to mention these points.
>
> Our empirical results demonstrate some potential use cases of this result, including analyzing 46-dimensional time series data ([new Fig 6](https://i.imgur.com/gVnReJj.png)) and solving control problems ([new Fig 3](https://i.imgur.com/BMAg3yU.png)).
>
> > calculation of the conditional probability given certain states’ representations is not a difficult task
>
> We agree that calculating conditional probabilities of Gaussian Markov chains is easy. We contend that calculating conditional probabilities of arbitrary time series data is very challenging (see, e.g., [Fig 4](https://i.imgur.com/XgDrGOu.png)). The contribution of this paper is to show how to learn representations of time series data to reduce this challenging problem to an easy problem. We have revised the start of Sec. 4 to mention these points.
>
> For example, consider the problem of inferring a sequence of states (i.e., a plan) between an initial state and a final state. Such planning problems typically require solving a combinatorial optimization problem (e.g., [Williams '17](https://ieeexplore.ieee.org/abstract/document/7989202), [Fang '23](https://arxiv.org/abs/2210.06601)). Even among prior learning-based methods that acquire compact representations, it can be unclear how to _efficiently_ perform planning over the space of representations.  In contrast, our work shows how to learn representations s.t. the planning problem becomes easy, reducing it to a simple matrix inversion (Sec. 4.4), avoiding the need for combinatorial optimization.
>
> > Comparison of PILCO and Gaussian processes for data-efficient learning in robotics and control
>
> We have added a discussion of these related works to the revised paper. These methods are similar to ours in that they aim to build a probabilistic model of time series data. Unlike GPs, our method avoids having to invert a matrix whose size depends on the number of data points.  A second difference is the aims of the papers: while those prior papers aim to learn reward-maximizing policies, our aim is to infer likelihood-maximizing sequences of observation representations. It's worth noting that the representation method we use (contrastive learning) has already seen widespread adoption in audio [e.g., [Oord '18](https://arxiv.org/abs/1807.03748)], computer vision [e.g., [Sermanet '18](https://arxiv.org/abs/1704.06888)], NLP [e.g., [Mikolov '13](https://arxiv.org/abs/1301.3781)], and reinforcement learning [e.g., [Laskin '20](https://proceedings.mlr.press/v119/laskin20a/laskin20a.pdf)], so providing a theoretically-grounded way of using these representations for inference may open new avenues for research.

---

> > ### Author Response · Authors · 2023-11-20
> > **Do the revisions address the concerns?**
> >
> > Dear Reviewer,
> >
> > Do the new revisions described above fully address the concerns about the paper? These revisions are targeted at addressing the excellent feedback raised in the review (e.g., clarifying the significance of the mathematical results). They also include new empirical experiments.
> >
> > Kind regards,
> >
> > The Authors

---

### Official Review · Reviewer_wbK6 · 2023-11-01

**Soundness:** 3 good
**Presentation:** 3 good
**Contribution:** 3 good
**Rating:** 5
**Confidence:** 2

**Summary:**

The authors analyze time-constravie learning methods (Info NCE with L2 regularization).
*  1) They showed learned representations take isotropic Gaussian distributions.
*  2) Under Assumption 1, they show that the conditional distribution for representation takes a closed-form solution, Gaussian distribution (Lemma 4.1.)
*  3) They showed that inference over representations can be similarly done in Sections 4.3 and 4.4.

**Strengths:**

## Presetation

* The question they address is essential.
* The writing way is overall clear.
* The analysis looks correct.

## Contribution

* The author asserts that the current body of literature on time-contrastive learning fails to elucidate the utility of acquired representations for the purpose of planning. If this assertion holds, I believe this paper's contribution carries substantial significance.

* The author contends that their approach is computationally more straightforward than sequential reconstruction-based methods. I agree with this assertion.

**Weaknesses:**

Note I clarify that this area is outside of my expertise. And, I am willing to change my score.


* There is a lack of comprehensive experimentation in Section 5.  I acknowledge the author's emphasis on the theoretical aspect of their work and  I highly value theoretical contributions. From what I gather, the author asserts that their work simplifies the planning process by deriving the analytical Gaussian form. I agree with this statement. However, at present, it remains challenging to envision how this method can offer practical utility in moderately complex environments and tasks. It raises questions about its direct applicability and the potential necessity of heuristics. The inclusion of empirical evidence would significantly strengthen the case for acceptance. Currently, I find myself in a somewhat undecided position.

* The relationship between Section 4 (inference over representations) and its applicability to specific tasks remains somewhat unclear. In RL, particularly in control tasks, our primary objective is to acquire the optimal policy, and any guarantees should pertain to the performance of this learned policy. It appears that this paper intends to convey the idea that learned representations can be useful for "inference over representations." However, it raises questions about whether "inference over representations" serves as the ultimate goal when analyzing data or if there are additional, more concrete objectives we should aim to address. Is there an opportunity to provide a more theoretically rigorous framework that aligns with the typical goals we seek to achieve?

**Questions:**

* Section 3: The paragraph titled "Our analysis will also look..." could benefit from substantial improvement. Currently, critical information resides in the Appendix, making it challenging to grasp the content of this section without consulting the Appendix. Ideally, drafts should be made to ensure that readers can comprehend the core content without the need for constant reference to the Appendix.

* Section 3: As a related suggestion, it would be advantageous to formalize the statements in this section within theorem/lemma/proposition frameworks, similar to the structure used in Lemma 4.1.

* In (4), there is no $c/(c+1)$?

* When l encounter with the term "planning," I initially inferred that it referred to the process of deriving an optimal policy in a control task. Is the current use of the term "planning" widely accepted within the community? In my perspective, the term "inference" appears to be a more precise descriptor.

* Is the primary assertion in Section 3, which states "learned representations follow isotropic Gaussian distributions," a novel contribution? The current phrasing does not distinctly convey what aspects are original. For instance, the reference to "supported by analysis based on Wang and Isola" is somewhat ambiguous. Does this imply that such analysis has been previously conducted in their work?

---

> ### Author Response · Authors · 2023-11-17
> **Author response: new experiments and revisions**
>
> Dear Reviewer,
>
> Thank you for the detailed feedback. It seems like the main suggestion is to add more experiments, which we have done through new experiments on higher-dimensional tasks (up to 46-dimensional) and by demonstrating that the inferred representations are useful for control. **Together with the revisions (red text in new PDF) and clarifications discussed below, does this fully address the reviewer's concerns about the paper?** We look forward to continuing the discussion!
>
> > The inclusion of empirical evidence would significantly strengthen the case for acceptance.
>
> As suggested, we have added 4 new figures with empirical results to validate our theory:
> * [Fig 3 (left)](https://i.imgur.com/BMAg3yU.png): We show how the analysis in Sec. 4 can be used to solve a maze. After learning representations, we used the closed form expression to infer the representations of intermediate states. This result shows how the representations effectively warp the state space so that linear motion in representations (i.e., $A \psi$) corresponds to fairly complex motion in the state space (i.e., navigating through a maze).
> * [Fig 3 (right)](https://i.imgur.com/BMAg3yU.png): Using this same maze, we show that the inferred waypoints can be used for control (details below).
> * [Fig 4](https://i.imgur.com/XgDrGOu.png): We apply the representations to a 39-dimensional robotic door opening task. Compared to a PCA baseline, linearly interpolated waypoints are a better fit for the true data waypoints.
> * [Fig 6](https://i.imgur.com/gVnReJj.png): We apply the representations to a 46-dimensional robotic hammering task. Compared to a PCA baseline, linearly interpolated waypoints are a better fit for the true data waypoints.
>
> > Evidence that planning by waypoint inference enables long-horizon control:
>
> To show waypoint inference is useful for control, we have added an experiment in a 2d maze environment ([Fig 3](https://i.imgur.com/BMAg3yU.png)). We define a simple proportional controller for navigating between nearby states; as expected ([blue line in Fig 3](https://i.imgur.com/BMAg3yU.png)), this proportional controller fails to reach distant goals. Using the proposed inference over representations, we infer waypoint representations, retrieve waypoints states (nearest neighbor), and iteratively have the proportional controller navigate to the waypoint states. The resulting success rates ([orange line in Fig 3](https://i.imgur.com/BMAg3yU.png)) are up to 4.5x higher (19% -> 87%).
>
> > "Our analysis will also look…": information resides in the Appendix
>
> We have added another half paragraph here to explain the intuition behind the contents in the Appendix. We welcome additional suggestions for details that should be moved to the main text.
> > Is the primary assertion in Section 3, which states "learned representations follow isotropic Gaussian distributions," a novel contribution?
>
> This assertion is a very minor extension of prior work, which is why we've placed it in the preliminaries section. We've added a note to explain this at the end of Section 3.
>
> > For instance, the reference to "supported by analysis based on Wang and Isola" is somewhat ambiguous. Does this imply that such analysis has been previously conducted in their work?
>
> We have revised this sentence to clarify the relationship with prior work. Prior work provides some theoretical intuition for why Assumption 2 should hold. For clarity, the Appendix explains why this intuition from prior work also holds for Gaussian distributions, rather than von Mises Fisher distributions.
>
> > Section 3: it would be advantageous to formalize the statements in this section within theorem/lemma/proposition frameworks, similar to the structure used in Lemma 4.1.
>
> We have reformatted this section to typeset the Assumptions in the same manner as the Lemmas in the paper. Our motivation for not phrasing Assumption 1 and Assumption 2 as "assumptions," rather than theorems, is that we want to be transparent about the assumptions behind our method. While it is true that both these assumptions are guaranteed to hold in certain settings, real world settings may violate these assumptions (e.g., because of sampling noise, function approximation error). We welcome other suggestions for making this section as precise as possible.
>
> > In (4), there is no $c / (c+1)$.
>
> We have fixed this typo.
>
> > Is the current use of the term "planning" widely accepted within the community?
>
> We have added a sentence to the introduction to explain that we will use planning to refer to an inference problem, rather than an optimal control problem. We provide citations there to prior work that takes a similar perspective.

---

> > ### Author Response · Authors · 2023-11-20
> > **Do the new experiments and revisions address the concerns?**
> >
> > Dear Reviewer,
> >
> > Do the new experiments and revisions described above fully address the concerns about the paper? We believe that these changes, motivated by the excellent feedback, further strengthens the paper. While the time remaining in the review period is limited, we would be happy to try to run additional experiments or make additional revisions.
> >
> > Kind regards,
> >
> > The Authors

---

> > ### Comment · Reviewer_wbK6 · 2023-11-21
> >
> > Thank you for your response. I will take it into account in the discussion phase.

---

### Meta-Review · Area_Chair_CTBJ · 2023-12-12

**Metareview:**

This paper presents a contrastive learning method for time series, which can be used for sequential decision-making.

Most of the reviewers express their concern about the applicability of this paper due to the lack of comprehensive experiments. Although the authors revised their paper to include several new numerical experiments, the effectiveness of the experiments has not been examined. Moreover, as pointed out by Reviewer wbK6, it remains unanswered whether their algorithms are useful to improve algorithms in RL.

Considering these unresolved issues, my recommendation is to reject the paper.

**Justification For Why Not Higher Score:**

See the above comments for the reason for rejection.

**Justification For Why Not Lower Score:**

N/A

---

### Decision · Program_Chairs · 2024-01-16

Reject